# Pathogenicity, Host Resistance, and Genetic Diversity of *Fusarium* Species under Controlled Conditions from Soybean in Canada

**DOI:** 10.3390/jof10050303

**Published:** 2024-04-23

**Authors:** Longfei Wu, Sheau-Fang Hwang, Stephen E. Strelkov, Rudolph Fredua-Agyeman, Sang-Heon Oh, Richard R. Bélanger, Owen Wally, Yong-Min Kim

**Affiliations:** 1Department of Agricultural, Food and Nutritional Science, University of Alberta, Edmonton, AB T6G 2P5, Canada; longfei@ualberta.ca (L.W.); sh20@ualberta.ca (S.-F.H.); strelkov@ualberta.ca (S.E.S.); freduaag@ualberta.ca (R.F.-A.); sangheo1@ualberta.ca (S.-H.O.); 2Centre de Recherche en Innovation des Végétaux, Université Laval, Québec, QC G1V 0A6, Canada; richard.belanger@fsaa.ulaval.ca; 3Harrow Research and Development Centre, Agriculture and Agri-Food Canada, Harrow, ON N0R 1G0, Canada; owen.wally@agr.gc.ca; 4Brandon Research and Development Centre, Agriculture and Agri-Food Canada, Brandon, MB R7C 5Y3, Canada

**Keywords:** *Fusarium* spp., genetic diversity, pathogenicity, phylogenetic analyses, resistance, root rot, Sanger sequencing, soybean

## Abstract

*Fusarium* spp. are commonly associated with the root rot complex of soybean (*Glycine max*). Previous surveys identified six common *Fusarium* species from Manitoba, including *F. oxysporum*, *F. redolens*, *F. graminearum*, *F. solani*, *F. avenaceum*, and *F. acuminatum*. This study aimed to determine their pathogenicity, assess host resistance, and evaluate the genetic diversity of *Fusarium* spp. isolated from Canada. The pathogenicity of these species was tested on two soybean cultivars, ‘Akras’ (moderately resistant) and ‘B150Y1′ (susceptible), under greenhouse conditions. The aggressiveness of the fungal isolates varied, with root rot severities ranging from 1.5 to 3.3 on a 0–4 scale. Subsequently, the six species were used to screen a panel of 20 Canadian soybean cultivars for resistance in a greenhouse. Cluster and principal component analyses were conducted based on the same traits used in the pathogenicity study. Two cultivars, ‘P15T46R2′ and ‘B150Y1′, were consistently found to be tolerant to *F. oxysporum*, *F. redolens*, *F. graminearum*, and *F. solani*. To investigate the incidence and prevalence of *Fusarium* spp. in Canada, fungi were isolated from 106 soybean fields surveyed across Manitoba, Saskatchewan, Ontario, and Quebec. Eighty-three *Fusarium* isolates were evaluated based on morphology and with multiple PCR primers, and phylogenetic analyses indicated their diversity across the major soybean production regions of Canada. Overall, this study contributes valuable insights into host resistance and the pathogenicity and genetic diversity of *Fusarium* spp. in Canadian soybean fields.

## 1. Introduction

Soybean (*Glycine max* L.) is one of the most important legume crops worldwide, known as one of the few ‘complete protein’ sources with all nine human-essential amino acids. Cultivation is mainly centered in North and South America, comprising 80% of total world production [1,2,3]. In Canada, soybean production has increased over the past decades, and it has become a crucial crop for human and livestock consumption as well as global exportation. Starting in Ontario and expanding to the Prairie provinces, soybean ranks behind only wheat, canola, and barley in acreage [4,5]. In 2022, 1.6 million ha of soybeans were grown in eastern Canada (mainly in Ontario and Quebec), yielding 5.1 million tonnes, with an additional 0.5 million ha (1.4 million tonnes) grown in western Canada (primarily Manitoba and Saskatchewan). The value of the crop in Canada was estimated at more than USD $33 billion [5,6].

Despite this rapid expansion, several biotic constraints continue to be a major limitation to soybean production. To date, over 300 diseases have been reported in soybean, causing yield losses averaging 11% [7]. A soybean survey conducted in the USA and Ontario found 23 specific diseases from 2015 to 2019, resulting in yield losses of 6–11% [8]. Addressing these biotic constraints is essential for enhancing soybean production, ensuring food security, and supporting farmers worldwide.

*Fusarium* spp. are significant soybean pathogens, causing root rot, wilt, sudden death syndrome (SDS), seed decay, and seedling blight [9,10,11]. In some cases, soybean diseases are caused by a single *Fusarium* species, for instance, *Fusarium* wilt caused by *F. oxysporum* Schlecht [12,13,14]. Others involve multiple species, constituting a pathogen ‘complex’. The soybean root rot complex, widespread in North America in recent years, results from infection by multiple *Fusarium* spp. [9,10,15,16,17]. These species include *F. solani*, *F. oxysporum*, *F. acuminatum*, *F. avenaceum*, *F. cerealis*, *F. culmorum*, *F. equiseti*, *F. graminearum*, *F. poae*, *F. proliferatum*, *F. pseudograminearum*, *F. redolens*, *F. sporotrichioides*, *F. fujikuroi*, *F. incarnatum-equiseti*, *F. tricinctum*, *F. semitectum*, *F. armeniacum*, *F. commune*, and *F. verticillioides* [9,10,15,16,18,19,20,21,22,23]. While *F. solani* and *F. oxysporum* were initially considered the dominant species in the soybean root rot complex, the composition of *Fusarium* spp. and their prevalence varied across different geographic regions [9,24,25]. Zhao et al. [19] reported that *F. proliferatum* is the most virulent species in soybean in Hubei, China. In contrast, in Sichuan province, the most aggressive species were *F. oxysporum*, *F. equiseti*, and *F. graminearum* [26]. At the reproductive stage of the crop, *F. acuminatum*, *F. graminearum*, and *F. solani* were more prevalent than other *Fusarium* spp. in Iowa [10]. Zhang et al. [27] reported eight *Fusarium* spp. associated with soybean root rot in Ontario, with the most aggressive species identified as *F. graminearum*. Moreover, *F. oxysporum* and *F. acuminatum* were the dominant species in surveys of soybean in Manitoba and Alberta, respectively [9,14].

Given the host-specificity and genetic diversity found within *Fusarium* spp., these can be further subdivided into *formae speciales* and/or races [14]. For example, within the *Fusarium solani* species complex (FSSC), *F. solani* is classified into 12 *formae speciales* and two races, distinguished by their host specificity and phylogenetic differences [28,29,30]. Four other members of the FSSC, *F. virguliforme*, *F. tucumaniae*, *F. brasiliense*, and *F. cuneirostrum*, serve as the primary causal agents of SDS [28]. However, there are instances where host-specific *formae speciales* have been observed to infect different hosts, underscoring the uncertainty and complexity involved in the classification of these special forms [31]. *Fusarium* spp. can also be categorized according to sexual compatibility, which grouped the members of the FSSC into seven distinct biological species (mating populations I–VII) [32]. Taking the *Fusarium oxysporum* species complex (FOSC) as another example, its *formae speciales* and races have been documented as pathogenic on over 150 hosts, with *F. oxysporum* f. sp. *vasinfectum* race 2 and *F. oxysporum* f. sp. *tracheiphilium* race 1 causing vascular wilt in soybean [33]. In western Canada, Hafez et al. [9] summarized four common species complexes: the *Fusarium tricinctum* species complex (FTSC), the *Fusarium incarnatum-equiseti* species complex (FIESC), the *Fusarium sambucinum* species complex (FSAMSC), and the FOSC, which included 10 of 12 *Fusarium* spp. identified in soybean and cereal crops in Manitoba. The remaining *Fusarium* spp., *F. solani* and *F. redolens*, belong to the FSSC and *Fusarium redolens* species complex (FRSC), respectively [34].

Generally, *Fusarium* spp. are classified based on various morphological characters, including colony features and pigmentation in different media, as well as the appearance, size, and presence of the three spore types: microconidia, macroconidia, and chlamydospores [35]. In addition to spores, mycelium and mycelial fragments may also cause infections [36,37,38,39]. The morphology, ecology, physiology, and even genetic traits of *Fusarium* spp. often exhibit variations across different studies [14,35,40,41]. Currently, the most common and effective approach for identifying *Fusarium* spp. in soybean relies on the DNA Sanger sequencing of PCR amplification products obtained with primers for specific genes or genomic regions. This method offers rapid, precise, sensitive, and convenient results [9,42]. The internal transcribed spacer (ITS) region of nuclear-encoded ribosomal DNA (ITS rDNA) is commonly used to generate primers to identify plant pathogens [43,44]. Both the ITS1/ITS4 and ITS5/ITS4 primer sets are popular for the identification of *Fusarium* spp. in various crops [14,19,45,46,47]. Additionally, numerous other primers targeting genes within the *Fusarium* genus have been widely applied in DNA sequence analysis. These alternate targets include the translation elongation factor 1-alpha (*TEF1α*) genes, mating type locus genes, phosphate permease gene, and beta-tubulin gene, as well as the largest and second largest subunit nucleotide sequences of RNA polymerase II, RPB1, and RPB2, respectively [9,48,49]. Furthermore, the most up to date *Fusarium* database, *FUSARIUM*-ID v.3.0, primarily identifies *Fusarium* spp. based on *TEF1α* sequences, complemented with some other loci, including RPB1, RPB2, and ITS [34]. At present, molecular methods with multiple primers, combined with the evaluation of morphological characters, are necessary to differentiate *Fusarium* spp. and analyze their phylogenetic relationships [19,26,50,51,52].

*Fusarium* spp. have a wide host range, infecting various crops such as cereals, soybean, other legume crops, canola, and corn [53,54,55,56]. Notably, *F. graminearum*, known for causing severe *Fusarium* head blight (FHB) in cereal crops, also exhibits high aggressiveness towards soybean [9,57,58]. Cross-pathogenicity among different crops can limit the efficacy of crop rotation in controlling *Fusarium* diseases [9,54]. As such, seed treatments are widely applied in North America to enhance seedling emergence and provide protection against soilborne pathogens on soybean [59,60,61]. Studies of fungicides and biocontrol agents as soybean seed treatments have also reported potential efficacy against single species of *Fusarium* under controlled conditions. The effectiveness of these treatments, however, remains limited in field trials conducted in natural environments [25,62,63].

The most promising strategy to manage soybean diseases caused by *Fusarium* spp. lies in the selection and breeding of resistant cultivars. However, globally available commercial cultivars with complete resistance are yet to be developed [27,64,65]. Incomplete resistance, controlled by polygenetic loci [66,67,68,69], is influenced by the interaction between the cultivar and the environment [70,71,72]. Cultivar screening and the detection of resistance to SDS caused by *F. virguliforme* in soybean have been reported in the United States [58]. Compared with complete resistance controlled by dominant genes, partial resistance is usually not race or species specific [73]. Considering the involvement of multiple *Fusarium* spp. in the development of Fusarium root rot (FRR) in soybean, broad-spectrum resistance (BSR) is a desirable trait for minimizing the risk of this disease. Tolerance or incomplete resistance to virulent isolates of *Fusarium* have also been identified based on reduced disease severity and other agronomic traits [27,62,74]. However, broad-spectrum resistance to *Fusarium* spp. has not been evaluated in Manitoba or other regions of Canada.

To improve the management of *Fusarium* diseases in Canadian soybean cultivation, it is essential to understand the species diversity, distribution, and pathogenicity of *Fusarium* spp., as well as to assess soybean cultivar resistance in the major production areas. This study aims to achieve three primary objectives: (1) to evaluate the pathogenicity of six common *Fusarium* spp. on soybean under controlled conditions; (2) to assess resistance/tolerance to these six common *Fusarium* spp. in a selection of 20 soybean cultivars; and (3) to investigate the genetic diversity and distribution of *Fusarium* spp. across the major soybean production regions of Canada.

## 2. Materials and Methods

### 2.1. Fungal Material and Inoculum Preparation

Twelve isolates representing the six most common *Fusarium* species (two isolates per species) in Manitoba as reported by Kim et al. [75], consisting of *F. oxysporum*, *F. redolens*, *F. graminearum*, *F. solani*, *F. avenaceum*, and *F. acuminatum*, were obtained from the culture collection of the Agriculture and Agri-Food Canada, Brandon Research and Development Centre, Brandon, Manitoba (Table 1). Monosporic purification and grain inoculum preparation for each isolate were accomplished according to a procedure modified from Chang et al. [15]. Briefly, each of the 12 isolates was subcultured from single-spores on PDA for 13–15 days at room temperature, and cultured on Difco™ potato-dextrose agar (PDA) (Becton, Dickinson and Company, Sparks, MD, USA) for one week at 25 °C in the dark. Eight kg (approximately 10 L) of wheat grain was soaked in distilled water overnight, and the soaked grain was transferred to an autoclave bag (each bag containing 600 mL of soaked grains) and autoclaved at 121 °C for 90 min. After cooling, the grain was inoculated with 10 pieces of 7-day-old mycelial plugs (5 mm × 5 mm) excised from the PDA colonies of each fungus and incubated at 25 °C in the dark for 2 weeks. The infested wheat grain was dried and ground into a powder (particle sizes between 0.25 and 1 mm). A final volume of 3 L dry grain inoculum was obtained for each isolate.

### 2.2. Pathogenicity Test

Two soybean cultivars, ‘Akras’ and ‘B150Y1′, were selected to evaluate the pathogenicity of the 12 fungal isolates. To avoid the potential interactions of other microorganisms with *Fusarium* spp., the potting mix (Sun Gro Horticulture Canada Ltd., Seba Beach, AB, USA) was autoclaved twice at 121 °C for 120 min. Plastic cups (473 mL vol.) were filled with sterilized soil mix, and a layer of grain inoculum powder (10 mL) was applied to each cup and covered with 1 cm of soil mix. Seven seeds of each soybean cultivar were sown into each cup and covered with an additional 1 cm of soil mix. The cups were then placed in a greenhouse maintained at 25 °C with natural light supplemented by artificial lighting (light/dark cycle of 18 h/6 h). Non-inoculated controls of each soybean cultivar received sterilized wheat grain powder that had not been inoculated with any of the *Fusarium* spp. The pathogenicity test was arranged in a randomized complete block design (RCBD) with five replicates and repeated twice.

### 2.3. Evaluation of Resistance

The more virulent isolate of each *Fusarium* spp. was selected based on the results of the pathogenicity test and applied to screen a germplasm collection of 20 soybean commercial cultivars, including ‘HS11Ry07’, ‘S15B4’, ‘P15T46R2’, ‘B150Y1’, ‘Williams’, ‘Akras’, ‘TH32004R2Y’, ‘AC Proteus’, ‘AAC Edward’, ‘AAC Springfield’, ‘OAC Prudence’, ‘Misty’, ‘NSC Reston’, ‘OAC Ayton’, ‘Bloomfield’, ‘AC Harmony’, ‘OAC Petrel’, ‘Mandor’, ‘OT15-02′, and ‘NSC Dauphin’. Seeds were provided by the University of Alberta, and Agriculture and Agri-Food Canada (Harrow Research and Development Centre, Morden Research and Development Centre, and Brandon Research and Development Centre). The same seeding and inoculation methods were used as described above for the pathogenicity test. To ensure adequate inoculum pressure, 15 mL of grain inoculum was applied to each cup. All 20 cultivars were inoculated with each of the six *Fusarium* spp. Non-inoculated controls, prepared as above, were also included for each cultivar. The experiment was arranged in an RCBD with five replicates and repeated twice.

### 2.4. Greenhouse Data Collection

Seedling emergence, plant height, root rot severity (RRS), and dry shoot and root weights were recorded in the pathogenicity testing and evaluation of host resistance. Emergence counts were conducted on the 7th, 14th, and 21st day after seeding. The plant height from stem to the top leaf was measured on the 14th day in centimeters. Symptoms on the soybean seedlings varied in response to the six *Fusarium* spp.; for instance, *F. avenaceum* caused reddish-brown discoloration on the roots, *F. solani* caused brown discoloration, while *F. oxysporum* attacked the root vascular system and caused a general tracheal mycosis. To enable comparisons, a 0–4 scale of RRS was applied as previously described in other studies [15,47,76], with minor modifications. Briefly, the extent of lesions and discoloration of the total root system was considered, together with the adverse effects of the damaged root system on overall plant health. Ratings consisted of 0 = no symptoms; 1 = limited visible lesions or discoloration on roots, with aboveground growth appearing normal; 2 = extended lesions or discoloration on roots, with slightly reduced aboveground growth; 3 = severe lesions or discoloration on roots, with severely reduced aboveground growth; and 4 = dead plant. This disease scale was applied to rate symptoms caused by each of the six *Fusarium* spp. evaluated (Figure 1). The 21-day-old seedlings from each replicate were collected after disease rating; their shoots and roots were separated, dried at 35 °C for 48 h, and weighed.

### 2.5. Fungal Isolation from Field Samples

Soybean root samples were collected from 106 fields in Manitoba (55 fields), Saskatchewan (18 fields), Ontario (30 fields), and Quebec (3 fields) during the 2022 growing season. Fifteen roots with symptoms of root rot were chosen from each field for pathogen isolation. Two symptomatic root pieces (3–5 mm long) were excised from the root tip and crown of each selected root, surface-sterilized in 1% NaOCl for 60 s, and then rinsed three times with sterilized water. The surface-sterilized root pieces were transferred to PDA containing 0.2 mg/mL streptomycin and incubated for 7 days at 24 °C. The cultures were examined and any colonies of *Fusarium* were transferred to water agar and incubated for 5 days. Afterward, a single hyphal tip was cut and transferred to PDA in Petri dishes for purification. The purified cultures were grouped and sub-grouped based on their morphological characteristics, including color, mycelium type, and released pigment on both sides of the Petri dishes [35].

### 2.6. DNA Extraction, PCR Amplification, and Sanger Sequencing

The purified cultures were grouped based on their morphology on PDA and one isolate was randomly selected from each group for molecular identification. A total of 336 isolates from 89 fields and six reference isolates of *Fusarium* spp. were used to extract genomic DNA using a modified CTAB method following O’Donnell et al. [77]. Approximately 100 mg of the mycelium was scraped from the surface of a 10-day-old colony grown on PDA and transferred to a 1.5 mL microcentrifuge tube. The samples were flash-frozen in liquid nitrogen and mechanically homogenized using a TissueLyser II (Qiagen, Hilden, Germany) at 25 Hz with 3 mm beads for 1 min. Seven hundred µL of CTAB buffer (Teknova, Hollister, CA, USA) was added to each 1.5 mL microcentrifuge tube and incubated at 65 °C for 30 min. A 500 µL aliquot of phenol:chloroform:isoamylalcohol (25:24:1) (Invitrogen, Waltham, MA, USA) was added to each extraction tube and centrifuged at 1107× *g* (3000 rpm) for 30 min. The aqueous phase was transferred to a new microcentrifuge tube and mixed with an equal volume of isopropanol, incubated at −20 °C for 1 h, and then centrifuged for 30 min at 1107× *g* (3000 rpm). The supernatant was decanted, and the DNA pellet was washed twice with 70% ethanol and allowed to dry overnight at room temperature. The DNA pellet was dissolved in 40 µL TE buffer (10 mM Tris-HCl, 1 mM EDTA), the DNA concentration was estimated using a Nanodrop ND2000 (Thermo Fisher Scientific, Wilmington, CA, USA), and diluted to a final concentration of 50 ng DNA/µL in distilled water.

The primer sets ITS4/5 (5′GGAAGTAAAAGTCGTAACAAGG 3′/5′TCCTCCGCTTATTGATATGC 3′; ITS region) [44] and T12 (5′AACATGCGTGAGATTGTAAGT 3′/5′TAGTGACCCTTGGCCCAGTTG 3′; beta-tubulin gene) [46,77] were used for PCR amplification and sequencing of each DNA sample. A subset of DNA samples identified as *Fusarium* spp. by ITS4/5 and T12 was further selected for species distinction with the primer set EF1/2 (5′-ATGGGTAAGGARGACAAGAC-3′/5′-GGARGTACCAGTSATCATG-3′; *TEF1α* gene) [78]. Amplifications were conducted in a 50 µL reaction volume, which included 1 ng/μL DNA template (5 µL), 10× Go Taq buffer (10 µL), MgCl_2_ buffer (5 µL), sterile distilled water (26.75 µL), 2 mM dNTPs (1 µL), 1 μL each of forward and reverse primers (2.5 μM), and 0.25 μL of DNA Taq polymerase (Titanium Taq, Promega, Madison, CA, USA). The PCR conditions were set as follows: initial denaturation at 95 °C for 120 s; followed by 40 cycles at 95 °C for 45 s, annealing for 45 s, and extension at 72 °C for 1 min; and a final extension at 72 °C for 10 min followed by cooling at 4 °C until recovery of the samples. The annealing temperatures were 55 °C and 52 °C for the ITS4/5 and T12 beta-tubulin genes, respectively. A 2 µL aliquot from each PCR was examined for the presence/absence of amplicons by electrophoresis in 2% agarose. The amplicons in the remaining 48 µL volume were purified using a QIAquick PCR Purification Kit (Qiagen, Hilden, Germany) and sent for Sanger sequencing at the Molecular Biology Services Unit (MBSU) of the University of Alberta in Edmonton, AB. The sequences obtained were used in BLAST searches of the National Center for Biotechnology Information (NCBI) databases (https://blast.ncbi.nlm.nih.gov/Blast.cgi; accessed on 1 January 2024). The BLAST comparison was conducted in standard databases (nr etc.) with the organism of nucleotide collection (nr/nt) using highly similar sequences (megablast). The BLAST results were sorted by percent identity, of which the first result was used to identify the species of the DNA sample.

### 2.7. Phylogenetic Analysis

All 227 sequences amplified by ITS4/5 and T12, as well as 89 sequences generated by EF1/2 from the isolates obtained from the soybean root samples collected in 2022 and the six reference isolates, were edited using Bio-Edit software v. 7.2.5 [79], with manual adjustment. The six reference sequences that best matched *F. oxysporum*, *F. redolens*, *F. graminearum*, *F. solani*, *F. avenaceum*, and *F. acuminatum* were downloaded from GenBank. All the sequences identified as *Fusarium* spp. and the six reference sequences were included in a phylogenetic analysis by the maximum likelihood (ML) method using the Jukes Cantor model with default parameters in the CLC main Workbench v.23.0.3 (QIAGEN, Aarhus, Denmark). Bootstrap values (BV) (%) were calculated with 1000 replicates, and a phylogenetic tree was constructed with BV > 70. The phylogenetic tree was finalized by iTOL version 6.8.1 (https://itol.embl.de/; accessed on 12 January 2024).

### 2.8. Data Analysis

Analysis of variance (ANOVA) was conducted for all traits analyzed in the pathogenicity test and cultivar resistance evaluation trials in the greenhouse using R v. 4.2.0 [80]. As the experiments were arranged in a RCBD with five replicates, the experimental unit was defined as each plastic cup. The seven seedlings in each cup (whether germinated or non-germinated) were considered as pseudo-replicates. Thus, the germinated seedlings in each cup on the 7th, 14th, and 21st day after seeding were counted as “count1”, “count2”, “count3”. The plant height and RRS of each experimental unit were estimated as “Height” and “RRS”, respectively, based on the mean of the seven pseudo-replicates in each cup, of which non-germinated/non-emerged seeds or seedlings were regarded as dead (height = 0 cm; RRS = 4). The RRS of non-germinated/non-emerged seeds or seedlings for the non-inoculated controls was rated as 0, since the germination failure was not caused by pathogen infection. The total dry shoot and root weights (g) in each cup were denoted as “shoot” and “root”, respectively. The estimated mean of all traits for the two greenhouse experiments was calculated based on the five replicated experimental units. A least significant difference (LSD) at *p* < 0.05 was applied to compare the estimated means of all traits among different treatments using the R package “agricolae” version 1.3-5. Correlation coefficients among all the traits collected in the pathogenicity test were calculated with the R package “PerformanceAnalytics” version 2.0.4. The reduction percentage was calculated for cluster analysis using R v. 4.2.0 [80]. To demonstrate the levels of high tolerance, moderate tolerance, moderate susceptibility, and high susceptibility, the cluster group number was set as four. A principal components analysis (PCA) was carried out, estimating the reduction percentage of the three germination counts, height, shoot weight, and root weight, as well as the increase in RRS (DS) in R using the package “ggbiplot” version 0.55. For the *Fusarium* spp. isolation, the incidence of each species was calculated using the formula: Incidence % = (n/N) × 100, where n is the number of fields where the species was detected in each province, and N is the total field number in each province.

## 3. Results

### 3.1. Pathogenicity Test

Symptoms caused by the six *Fusarium* spp. varied and included rotting, girdling, and the development of brown sunken lesions (Figure 1). An analysis of variance indicated significant interactions between *Fusarium* spp. and the soybean cultivars for all the traits, while the repeat effect of two greenhouse experiments was not significant (Table 1). Consequently, data from the two repeated experiments were combined for all traits. Mean and LSD estimates for the six *Fusarium* spp. and non-inoculated control were calculated using ‘Akras’ and ‘B150Y1’ for all traits. Correlation analysis indicated highly significant correlation coefficients among all the tested traits (Appendix A). RRS, as a direct reflection of pathogen infection, was negatively correlated with the other traits, including the three emergence counts and their average, plant height, and dry shoot and root weights. The reduction in these growth parameters was considered together with RRS when evaluating the aggressiveness of the *Fusarium* species in this study.

Reductions in the emergence of the cultivar ‘B150Y1′ were not significant for any of the six *Fusarium* species or isolates tested except for *F. avenaceum* isolate 1. In contrast, significant reductions in emergence were observed for ‘Akras’ following inoculation with each of the isolates (Figure 2a). In general, ‘B150Y1′ had a higher emergence rate than ‘Akras’ under both inoculated and non-inoculated conditions. Differences between the two isolates of each species were significant only for *F. redolens*, *F. solani*, and *F. avenaceum* on the cultivar ‘Akras’. The greatest reduction in emergence (50%) was observed with *F. avenaceum* isolate 1.

A significant reduction in plant height was observed for ‘B150Y1′ inoculated with each of the 12 isolates, as well as for ‘Akras’ inoculated with *F. redolens* isolate 2, *F. graminearum* isolate 1, and both isolates of each of *F. solani*, *F. avenaceum*, and *F. acuminatum* (Figure 2b). Variance within each *Fusarium* species reached a significant level for *F. redolens*, *F. graminearum*, and *F. avenaceum* on ‘Akras’, while isolate effects within species were not significant for ‘B150Y1′. In addition, ‘B150Y1′ had relatively higher plant height than ‘Akras’. The lowest plant height was obtained for ‘Akras’ inoculated with *F. avenaceum* isolate 1, followed by the same host inoculated with *F. avenaceum* isolate 2 and *F. graminearum* isolate 1.

All isolates of the six *Fusarium* spp. caused significant increases in RRS relative to the non-inoculated controls for both cultivars (Figure 2c). In the case of *F. oxysporum*, RRS caused by isolate 2 was significantly greater than RRS caused by isolate 1. Similarly, *F. redolens* isolate 2 caused significantly more severe disease on ‘Akras’ than isolate 1, while disease severities on ‘B150Y1′ were similar for both isolates. On ‘Akras’, *F. solani* isolate 1 caused the highest RRS, but no significant differences were observed between the two host cultivars. *Fusarium graminearum* caused greater RRS on ‘Akras’ than on ‘B150Y1′, with both isolates of this species inducing similar levels of disease. Isolate 1 of *F. avenaceum* caused higher RRS than isolate 2 on both cultivars, while *F. acuminatum* isolate 2 was more virulent than isolate 1 on these hosts.

Reductions in shoot dry weight were significant following inoculation with each of the isolates and ranged from 24.0% for ‘B150Y1′ in response to inoculation with *F. graminearum* isolate 2 to 70.8% for ‘Akras’ in response to *F. avenaceum* isolate 1 (Figure 2d). Similarly, shoot weight reductions were significant for all the isolates and both soybean cultivars. In general, the root dry weight of ‘B150Y1′ was greater than for ‘Akras’. Reductions in root dry weight following the inoculation of both cultivars were mostly significant (Figure 2e), with the only exceptions found for *F. avenaceum* isolate 1 on ‘B150Y1′, *F. solani* isolate 2 on ‘Akras’, and *F. acuminatum* isolate 1 and isolate 2 on ‘Akras’. Distinct differences between the root weight reductions caused by the two isolates of each species were detected only in *F. redolens*, *F. solani*, and *F. avenaceum*.

In the pathogenicity study, the aggressiveness of each isolate was consistent across all measured traits including RRS and reductions in emergence, plant height, and dry shoot and root weights. The most aggressive isolate of each species, identified as *F. oxysporum* isolate 2, *F. redolens* isolate 2, *F. solani* isolate 1, *F. graminearum* isolate 2, *F. acuminatum* isolate 2, and *F. avenaceum* isolate 1 based on these parameters, was selected to evaluate cultivar resistance.

### 3.2. Cultivar Resistance Evaluation

Both greenhouse experiments yielded corresponding results for all measured variables, with a non-significant repeat effect (*p* > 0.05) (Table 2). Cultivar effect, *Fusarium* spp. effect, and their interaction were all found to be significant for all the traits (*p* < 0.05) (Table 2). As such, the reaction of each of the 20 soybean cultivars was evaluated in response to each of the six *Fusarium* spp. Notably, significant differences among the soybean cultivars were observed for all traits in the non-inoculated control, with ranges of 3.9–7.0 for the three germination counts (Count1, Count2, Count3), 5.4 cm–17.1 cm for plant height, 0.9 g–1.5 g for dry shoot weight, and 0.2g–0.4 g for dry root weight. As expected, however, the non-inoculated controls of all 20 cultivars remained completely healthy, with a score of 0 for RRS (Appendix A). Consequently, the disease reactions for the six *Fusarium* spp. were evaluated using percentage reduction for Count1, Count2, Count3, Height, Shoot, and Root, as well as RRS increase.

*Fusarium avenaceum* caused the greatest average reduction in emergence (94.1%), while the maximum average height reduction was observed for *F. oxysporum* (99.5%). For shoot and root loss, *F. oxysporum* had the most pronounced effects, with reductions of 97.6% and 98.7%, respectively. Regarding RRS, inoculation with *F. oxysporum* and *F. avenaceum* caused the greatest severity scores for RRS (3.9). In contrast, *F. solani* had the lowest impact on all the traits compared with the other five *Fusarium* spp.

Principal component analysis with four clusters revealed a positive correlation among variables and cultivar reactions following inoculation with *F. graminearum*, *F. avenaceum*, *F. acuminatum*, *F. oxysporum*, *F. redolens*, and *F. solani* (Figure 3). Notably, four cultivars, ‘B150Y1′, ‘P15T46R2′, ‘Misty’, and ‘Mandor’, clustered in the high-tolerance group against *F. graminearum*. Against *F. avenaceum*, the most tolerant cultivars were ‘OAC Prudence’, ‘NSC Reston’, ‘OAC Ayton’, and ‘AC Harmony’. The cultivars ‘S15B4′, ‘OT15-2′, and ‘Mandor’ displayed the greatest tolerance to *F. acuminatum,* and ‘P15T46R2′, ‘B150Y1′, ‘Williams’, and ‘AAC Edward’ were most tolerant to *F. oxysporum*. Additionally, the cultivars most tolerant to *F. redolens* were ‘S15B4′, ‘P15T46R2′, ‘B150Y1′, ‘TH32004R2Y’, ‘AC Proteus’, and ‘OAC Petrel’. The cultivars, ‘HS11Ry07′, ‘S15B4′, ‘P15T46R2′, ‘B150Y1′, ‘Williams’, and ‘OAC Prudence’ showed the greatest tolerance to *F. solani*. Overall, the soybean cultivars ‘P15T46R2′ and ‘B150Y1′ displayed suppression against *F. graminearum*, *F. oxysporum*, *F. redolens*, and *F. solani*.

### 3.3. Fusarium spp. Identification

A total of 983 purified isolates were obtained from symptomatic root samples and separated into four morphological groups (“Red”, “White”, “Purple”, and “Other”) and nine subgroups (“Red1”, “Red2”, “Red3”, “Red4”, “White1”, “White2”, “White4”, “Purple1”, and “Slimy1”) (Figure 4). After filtering and grouping based on colony morphology, 336 isolates were selected for molecular identification. The primer sets ITS4/5 (ITS region), T12 (beta-tubulin gene), and EF1/2 (*TEF1α* gene) produced single bands of ~500 bp [47], ~580 bp [46], and ~690 bp [78], respectively, from the six reference isolates (Figure 5) and isolates collected in this study. Following the removal of non-*Fusarium* species and isolates with poor sequence quality, 221 isolates were confirmed as *Fusarium* spp., primarily based on the ITS4/5-amplified gene sequences complemented with the T12-amplified sequences.

Within the 221 *Fusarium* spp. isolates detected by ITS4/5 and T12, 33 (14.9%), 47 (21.2%), and 141 (63.8%) isolates had coincident identification, species conflict, and single primer identification, respectively. The inconsistency of sequence identification between the ITS region and the beta-tubulin gene was mainly observed in the species within the FTSC as well as those belonging to FEISC. Thus, 83 isolates and 6 reference isolates were further selected for confirmation by EF1/2 as a subgroup, consisting of 6 coincident isolates, 29 isolates with conflict, 48 isolates with a single result, as well as 6 reference *Fusairum* spp. (Appendix A). Based on the sequence blast of the *TEF1α* gene, 19 of 80 isolates exhibited coincidence with results based on the ITS region. For the comparison with 35 sequences of the beta-tubulin gene, 21 sequences were identified as the same *Fusarium* species by the *TEF1α* gene. In the identification conflicts between *F. avenaceum* and *F. acuminatum* with the primer sets ITS4/5 and T12, the species identifications obtained with EF1/2 were consistent with 17 of 20 sequences amplified by T12. Among conflicting isolates belonging to the FIESC, 13 of 14 were identified by the *TEF1α* gene as *F. avenaceum* while only one was *F. equiseti*. None of these matched the species identifications based on the ITS region or the beta-tubulin gene.

Overall, the main species identified with the EF1/2 primer set were *F. avenaceum* (39 isolates), *F. acuminatum* (25 isolates), *F. oxysporum* (10 isolates), and *F. redolens* (3 isolates). On the other hand, only one isolate each of *F. culmorum*, *F. equiseti*, *F. flocciferum*, *F. sporotrichioides*, *F. solani*, and *F. falciforme* were confirmed with these primers. Isolates identified as *F. avenaceum* in the current study had the greatest diversity in morphology, and were observed to occur in all the colony morphology subgroups. *Fusarium avenaceum* and *F. acuminatum* both occurred in the morphology subgroups “Red1” and “Red2”. Morphological variation was also detected in *F. oxysporum*. In contrast, *F. redolens* was only detected in the “White2” group. Isolates of *F. culmorum*, *F. flocciferum*, and *F. sporotrichioides*, all belonging to the FSSC, were found in the “Red4” morphology subgroup. All the major species were detected in both western and eastern Canada apart from *F. redolens*, which was identified only in Manitoba and Saskatchewan. While the current study identified other *Fusarium* spp. in the FSSC, the reference species *F. graminearum* was not identified in this soybean survey.

### 3.4. Phylogenetic Analysis

The ML phylogenetic analysis based on the *TEF1*α-amplified sequences clearly segregated all the identified *Fusarium* spp. into seven clades with bootstrap values ranging from 89 to 100%, except for NSRR22_062_red1_SK_*F. flocciferum*, which was grouped with isolates identified as *F. avenaceum* (Figure 6). The phylogenetic trees were rooted using sequences of all *F. avenaceum* and the single *F. flocciferum*, with the reference sequence of *F. avenaceum* in Clade1. Clade2 included all the isolates identified as *F. acuminatum* and its reference sequence, with a bootstrap value of 89%. Clade3 included one *F. solani*, one *F. falciforme*, and the *F. solani* reference sequence in the FSSC, with a bootstrap value of 99% [81]. NSRR22_230_white4_SK_*F. equiseti* was the only isolate in the FIESC and was classified as Clade4, distinguished from Clade5 with a 99% supporting value. The latter (Clade5) consisted of *Fusarium* spp. in the FSAMSC, including *F. culmorum*, *F. sporotrichiodes*, and the reference sequence of *F. graminearum* with a bootstrap value of 100. The three *F. redolens* isolates, along with the reference sequence, were strongly linked together as Clade6 (bootstrap value = 100%). Similarly, all 10 isolates of *F. oxysporum* and its reference sequence were clustered as Clade7 with a bootstrap value of 92%.

The phylogenetic tree generated by ITS4/5-amplified sequences generally included seven groups (Appendix A). However, species mixture was observed in groups 1, 3, and 6, corresponding to FTSC, FIESC, and FOSC. Meanwhile, the bootstrap values of nodes for grouping were relatively low. Considering the high failure rate of sequence BLAST and inconsistencies in species identification by ITS4/5, T12, and EF1/2 of isolates in morphology subgroup “White4”, only sequences identified as a species of FTSC by amplification of the beta-tubulin gene were analyzed for their phylogenetic relationships (Appendix A), of which, *F. avenaceum* was significantly distinguished from *F. acuminatum*.

## 4. Discussion

Root rot in soybean is a global concern involving numerous soilborne pathogens, with *Fusarium* spp. found to be predominant in disease surveys conducted in eastern [74] and western Canada [9,14,16]. However, studies investigating the pathogenicity of *Fusarium* spp. in Canada, especially in the western Prairie region (Alberta, Saskatchewan, and Manitoba), are limited.

While several *Fusarium* spp., including *F. oxysporum* and *F. solani* [28,33], consist of distinct *formae speciales* or physiological races, these species are also reported to be pathogenic in multiple crops [28,57,82]. The *F. graminearum* species complex, which is the primary cause of Fusarium head blight in cereals such as wheat, barley, and other small grains [83], exhibits a wide host range that also extends to soybean, rice, and maize [58,84]. The pathogenicity and aggressiveness of the FGSC complex on cereals are well described and associated with the production of the mycotoxins nivalenol (NIV) and deoxynivalenol (DON) [85,86]. While *F. graminearum* is capable of infecting soybeans, there have not been any specific *formae speciales* reported for this crop. Assessing fungal aggressiveness based on a single parameter is likely insufficient, particularly for *Fusarium* species that are pathogenic but not specialized for soybeans. Thus, the evaluation in this study, not only of RRS but also of emergence, plant height, and dry shoot and root weights, provided a more comprehensive description of the aggressiveness of *Fusarium* spp. on soybean. Likewise, by examining multiple traits and incorporating principal components analysis, a more comprehensive assessment of the tolerance of various soybean cultivars to Fusarium root rot was achieved.

In Canada, Abdelmagid et al. [16] evaluated the pathogenicity of five *F. sporotrichioides* isolates from Manitoba, which caused up to 70% RRS and significant reductions in root and shoot lengths in soybean. The cross-pathogenicity of five *Fusarium* spp., including isolates of *F. cerealis*, *F. culmorum*, *F. graminearum*, and *F. sporotrichioides* collected in Manitoba, was also examined on soybean and wheat; the resulting RRS on soybean ranged from 1.89 to 3.33 on a 0–4 scale [9]. In Alberta, *F. proliferatum* was reported as the most aggressive species on soybean based on greenhouse trials, while other tested *Fusarium* spp. caused mild to moderate levels of disease [14]. In this study, the pathogenicity of six *Fusarium* species was compared on soybean, using isolates previously collected from Manitoba. These species included *F. oxysporum*, *F. graminearum*, *F. solani*, and *F. acuminatum*, which are common in North America, as well as *F. avenaceum*, which is also prevalent in Canada, and *F. redolens*, which was previously reported in the country [9,14]. The severity of root rot caused by these six species varied, with *F. oxysporum*, *F. avenaceum*, and *F. graminearum* causing the most severe root rot on the soybean cultivar ‘Akras’. Despite this variability, the root rot severities caused by the six *Fusarium* spp. tested in this study were generally greater than in previous reports [14,62,63], suggesting increased aggressiveness in these species. This trend of increasing virulence should be emphasized to farmers when implementing control measures for *Fusarium* root rot in soybean.

Several studies have detected horizontal resistance in soybean controlled by polygenetic loci against *Fusarium* spp. Acharya et al. [67] identified one major and one minor QTL on soybean chromosomes 8 and 6 controlling partial resistance to *F. graminearum*. Quantitative resistance was also detected against SDS on all 20 soybean chromosomes [66,68]. In this study, 20 commercial soybean cultivars were screened for resistance to virulent isolates of *F. oxysporum*, *F. graminearum*, *F. solani*, *F. acuminatum*, *F. avenaceum*, and *F. redolens* selected from the earlier pathogenicity test. The host reactions indicated varying degrees of tolerance. *Fusarium oxysporum* and *F. avenaceum* were identified as the most virulent species, while *F. solani* caused the lowest RRS, consistent with the findings of the pathogenicity testing. Complete resistance (RRS ≤ 1) was not observed in any of the 20 soybean cultivars evaluated. Moderate resistance was only observed against *F. redolens* and *F. solani* (1 ≤ RRS ≤ 2) (Appendix A). In this context, the level of tolerance was taken into consideration, evaluating all the traits investigated in this study. The six *Fusarium* spp. tested also had varying effects on germination counts, plant height, dry shoot weight, and dry root weight. In another study, 57 commercial soybean cultivars were evaluated against *F. oxysporum*, *F. graminearum*, *F. avenaceum*, and *F. tricinctum*, with resistance against all four species identified in the soybean cultivar ‘Maple Amber’, based on RRS rather than emergence, plant height, or dry root weight [74]. However, in that study, RRS ranged from 0.5 to 2.7 on a 0–4 scale, and reductions in emergence, plant height, and dry root weight were generally <50%.

Nyandoro et al. [62] evaluated the resistance of 12 soybean cultivars against *F. avenaceum* in greenhouse trials and found high RRS among cultivars (ranging from 2.6 to 3.4) and considerable reductions in emergence (26.7 to 75.5%). Given the variable performance of soybean cultivars across different trials, PCA was conducted in the current study to evaluate tolerance to *Fusarium* spp., taking into account RRS, germination counts, plant height, dry shoot weight, and dry root weight. Positive correlations were observed among RRS, reduction of emergence, decline in plant height, and weight loss of the shoot and root, illustrating the adverse effects of the six *Fusarium* spp. on the entire soybean plant. As the cultivars were clustered into four groups based on multiple parameters, the direct identification of high tolerance to infection was determined by selecting cultivars located in the circle farthest from the parameter arrowhead. Some tolerant cultivars were identified against each *Fusarium* spp., with broad-spectrum resistance detected in ‘P15T46R2′ and ‘B150Y1′. The cultivars were tolerant/partially resistant to *F. graminearum*, *F. oxysporum*, *F. redolens*, and *F. solani*. Similarly, in recent studies, broad-spectrum resistance to multiple races of *Phytophthora sojae* was identified in soybean genotypes carrying the *RpsX*, *Rps11*, *Rps12*, and *Rps13* resistance genes, suggesting a promising approach for control of this pathogen [87,88,89]. Broad-spectrum resistance was also evaluated for the management of soybean mosaic disease, providing information helpful for reducing viral infections and mitigating their impact on soybean [90,91]. Nevertheless, broad-spectrum resistance to *Fusarium* species and/or races in soybean still appears limited. Further study of ‘P15T46R2′ and ‘B150Y1′ may contribute to improved knowledge of the genetic control of host reactions to infection.

Numerous studies have highlighted the variability of *Fusarium* spp. associated with the soybean root rot complex across different environments, locations, and years [11,92,93,94]. In the current study, 10 different *Fusarium* spp. were identified from a large collection of symptomatic roots using a combination of morphological and molecular methods (Appendix A). All six species included in the pathogenicity test and cultivar evaluation trials were also recovered in the fungal isolation study, with *F. avenaceum*, *F. oxysporum*, *F. acuminatum*, and *F. redolens* as the major groups. The current study also isolated one culture of *F. equiseti*, commonly reported in northern California and South Dakota in the USA [95,96], as well as in Ontario, Alberta, and Manitoba [9,14,27]. However, species such as *F. poae*, *F. graminearum*, *F. solani*, *F. sporotrichioides*, *F. tricinctum*, *F. torulosum*, *F. commune*, and *F. proliferatum*, which were previously reported as predominant in Canada [9,14,27], were only sparsely identified or absent in this study. Notably, isolates of *F. redolens* displayed location-specific characteristics, consistent with previous reports [9,14]. The absence of *F. graminearum* in the current soybean survey suggests that it may not play a significant role as a major pathogen causing root rot disease in soybean, although it showed significant levels of pathogenicity in the pathogenicity test. Factors such as different methodologies, variations in field locations, soybean cultivars, or environmental conditions may have contributed to discrepancies.

Colony morphology played a crucial role in the primary grouping of isolates from the same field, allowing for the selection of 336 isolates from the initial 983 for molecular identification. All isolates identified as *F. redolens* with the primer sets ITS4/5 or EF1/2 were found in the morphology subgroup “White2”. Likewise, all isolates identified as *F. acuminatum* with EF1/2 and T12 were found in morphology subgroups “Red1” and “Red2”, characterized by small colonies containing short and dense mycelium with a light to dark reddish color. *Fusarium oxysporum* had three morphology types, including “White1”, “White4”, and “Purple1”, while the only *F. equiseti* isolate occurred in “White4”. The similarity in morphology between *F. oxysporum* and *F. equiseti* has also been reported in tomato in northeast India [97]. Within the “Red” group, however, the four subgroups (“Red1”, “Red2”, “Red3”, and “Red4”) failed to distinguish whether the isolates belonged to FTSC or FSAMSC. Additionally, *Fusarium avenaceum* showed the largest variability in morphology, and occurred with other *Fusarium* spp. in all nine morphology subgroups. Consequently, no morphologically unique species were found in the current study. Non-negligible differences in species identification by morphology and molecular phylogenetics have been reported in other recent studies, illustrating the need to integrate both morphological characteristics and molecular technologies in the identification of *Fusarium* spp. [98,99].

The sequencing of the DNA fragments amplified by PCR based on the ITS region and beta-tubulin gene identified 225 and 79 isolates belonging to *Fusarium* spp., respectively. Inconsistencies were frequently observed among species identified within the same *Fusarium* species complex, leading to the need for further confirmation by the sequencing of the *TEF1*α gene. For example, while five isolates were classified as either *F. incarnatum* or *F. equiseti* with the primer set ITS4/5, they were recognized as *F. flagelliforme* (also a member of FIESC) with the T12 primer set (beta-tubulin gene). Notably, *F. flagelliforme* was not detected using ITS4/5 in this study. Based on an analysis of the *TEF1*α gene, however, only one of these isolates was confirmed as *F. equiseti*, while the rest were identified as *F. avenaceum*. Chang et al. [26] demonstrated that variable identification of isolates across species complexes or among species within the *Fusarium* genus is common in studies with multiple primers, including those targeting the beta-tubulin and ITS regions. For instance, ITS sequences have been reported as unsuitable for distinguishing *F. equiseti* or *F. incarnatum* [45]. Similarly, the beta-tubulin sequence has been applied to identify species in the FIESC and *F. chlamydosporum* species complexes (FCSC) [98], but failed to distinguish *F. armeniacum*, *F. acuminatum*, *F. sporotrichioides*, and *F. langsethiae* [100]. While the *TEF1α* gene has been widely used to define species and reveal phylogenetic relationships within the genus *Fusarium* in the past, multiple primers have been more frequently employed in recent studies [9,19,51,101,102].

Phylogenetic analysis using the *TEF1*α sequence successfully distinguished the 10 identified *Fusarium* spp., except for *F. flocciferum*, which was grouped with 25 *F. avenaceum* isolates. Moreover, the phylogenetic tree generated based on the *TEF1*α gene sequences clearly showed the genetic distances within and among the species complexes, including the FTSC, FSSC, FIESC, FSAMSC, and FOSC. The classification among species or species complexes was strongly supported, with bootstrap values ranging from 89% to 100%. On the other hand, phylogenetic analysis based on the ITS region could generally distinguish complexes, such as FTSC, FOSC, FIESC, FSAMSC, and FSSC. However, only *F. incarnatum*-*equiseti*, *F. redolens*, and *F. solani* had strong bootstrap (≥70) support. Beta-tubulin sequences clearly distinguished *F. avenaceum* and *F. acuminatum* in the FTSC. O’Donnell et al. [102] concluded that the beta-tubulin gene is not universally informative within *Fusarium* and is suitable only for distinguishing *Fusarium* spp., forming part of the *F. solani* and *F. incarnatum-equiseti* species complexes. This study was the first to use the T12 primer set, specific for the beta-tubulin gene, to separate *Fusarium* spp. in FTSC. Hafez et al. [9] also generated a phylogenetic tree that clearly distinguished *Fusarium* spp. of the FTSC, FSAMSC, FOSC, FIESC, *F. redolens*, and *F. solani* collected from Carman and Melita, Manitoba. In a study investigating *Fusarium* isolates from central and southern Alberta, overlapping species were found in phylogenetic trees based on both the ITS and *TEF1*α sequences [14]. *Fusarium oxysporum* was reported to have extensive genetic variation [10] and morphological variability [103]. A whole genome sequence study demonstrated the large diversity in FOSC in Australia among the identified clades [94]. Overall, the *TEF1*α genes clearly distinguished *Fusarium* species and species complexes in the phylogenetic tree, demonstrating the accuracy of the BLAST sequence alignment. Phylogenetic analysis based solely on ITS4/5 was insufficient to classify the different *Fusarium* species complexes or a single species, while beta-tubulin clearly distinguished *Fusarium* spp. in the FTSC.

## 5. Conclusions

The evaluation of the pathogenicity of six *Fusarium* spp. on soybean indicated that *F. avenaceum* and *F. oxysporum* were the most strongly virulent, while *F. graminearum*, *F. acuminatum*, and *F. redolens* also caused significant levels of disease. In contrast, *F. solani* was weakly virulent, causing mild symptoms of *Fusarium* root rot in soybean. An assessment of the reaction of a suite of 20 soybean cultivars to inoculation with each of the *Fusarium* spp., based on RRS, emergence, plant height, and dry shoot and root weight, indicated that while no hosts were completely resistant, some cultivars showed partial resistance or tolerance to the disease. The soybean cultivars ‘P15T46R2′ and ‘B150Y1′ were consistently tolerant to *F. graminearum*, *F. oxysporum*, *F. redolens*, and *F. solani*, making them promising candidates for farmers seeking to minimize the risk of *Fusarium* root rot. Furthermore, this study provided valuable insights into the distribution and composition of *Fusarium* spp. in the major soybean production areas of Canada, updating and complementing existing information on the *Fusarium* root rot complex in soybean cultivation. These findings may help to guide the development of effective measures to mitigate the risk of *Fusarium* root rot of soybean.

## Figures and Tables

**Figure 1 jof-10-00303-f001:**
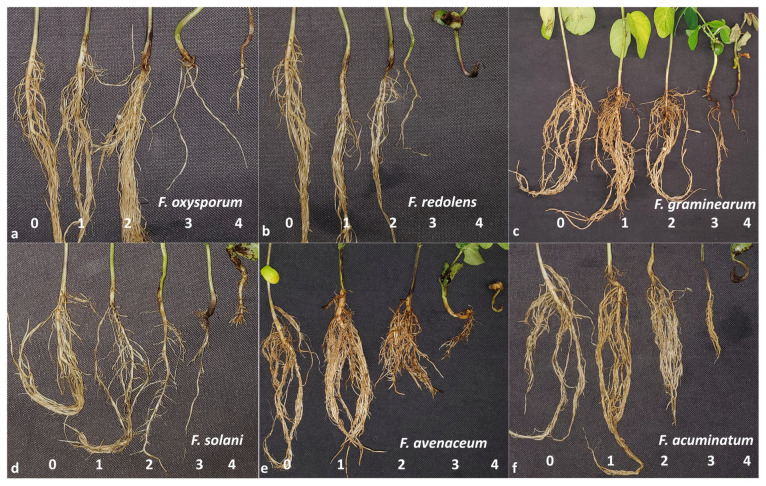
Root rot severity following inoculation of soybean with (**a**) *Fusarium oxysporum*, (**b**) *Fusarium redolens*, (**c**) *Fusarium graminearum*, (**d**) *Fusarium solani*, (**e**) *Fusarium avenaceum*, and (**f**) *Fusarium acuminatum*, assessed on a 0–4 scale following the methods outlined by Chang et al. [15] and Zhou et al. [14]. A brown discoloration was observed in soybean plants inoculated with all *Fusarium* spp. except for *F. graminearum* and *F. avenaceum*, which caused a dark reddish discoloration. Lesions between the root and stem were visible on plants inoculated with *F. oxysporum* and *F. redolens*, while *F. oxysporum*, *F. redolens*, and *F. acuminatum* caused a girdling at the base of the stem.

**Figure 2 jof-10-00303-f002:**
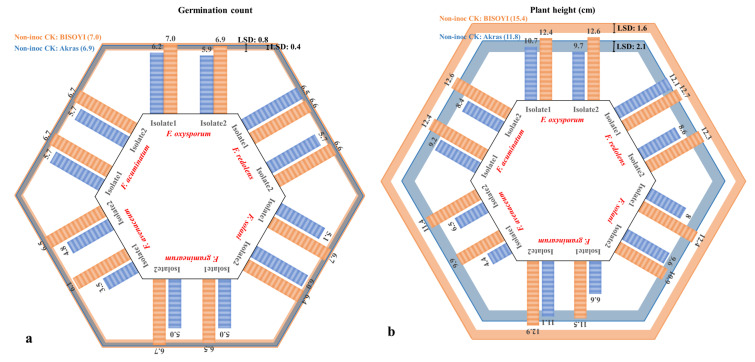
Effect of inoculation with each of the 12 *Fusarium* isolates on (**a**) germination count, (**b**) plant height, (**c**) root rot severity, and (**d**) dry shoot weight and (**e**) dry root weight of two soybean cultivars, ‘Akras’ and ‘B150Y1′. The orange and blue lines represent the estimated mean of the non-inoculated controls for ‘Akras’ and ‘B150Y1′, respectively. The bars indicate the values in response to the different fungal isolates. The least significant differences for ‘Akras’ and ‘B150Y1′ are also presented in the graphs, and indicate a significant effect among the 12 isolates as well as the six *Fusarium* species.

**Figure 3 jof-10-00303-f003:**
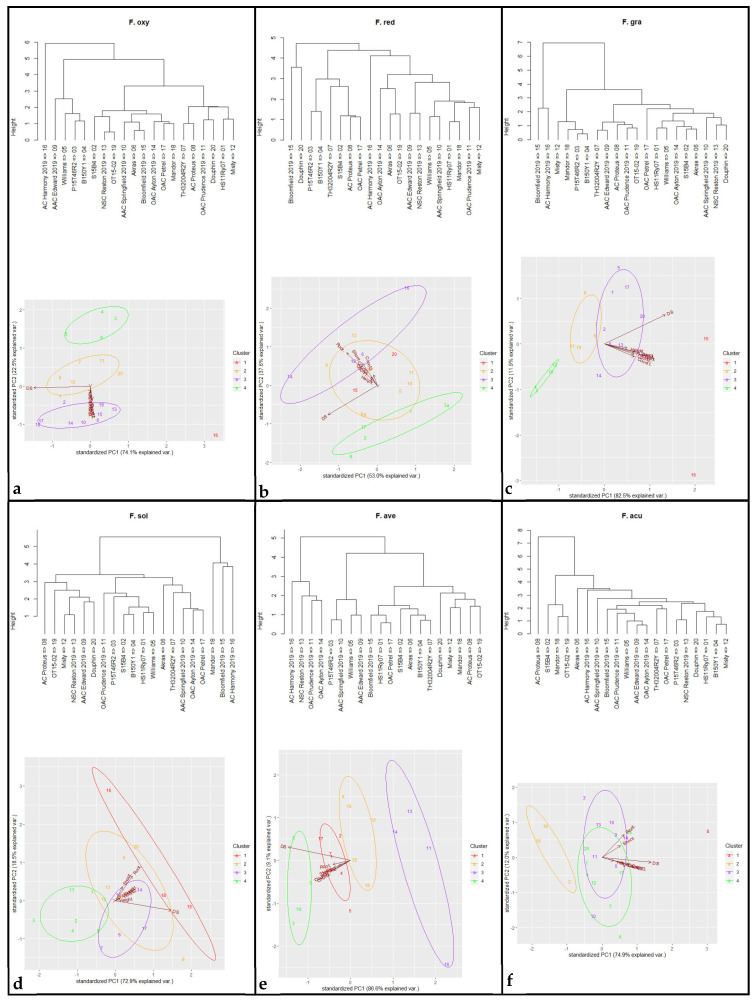
Cluster and principal component analyses of 20 soybean cultivars evaluated against six *Fusarium* spp., including (**a**) *Fusarium oxysporum*, (**b**) *Fusarium redolens*, (**c**) *Fusarium graminearum*, (**d**) *Fusarium solani*, (**e**) *Fusarium avenaceum*, and (**f**) *Fusarium acuminatum*. The evaluated traits included three emergence count (Count1, Count2 and Count3), plant height (Height), root rot disease severity (DS), dry shoot weight (Shoot) and dry root weight (Root). The hosts were divided into four groups (denoted by circles; high tolerance, moderate tolerance, moderate susceptibility, and high susceptibility) based on cluster analysis.

**Figure 4 jof-10-00303-f004:**
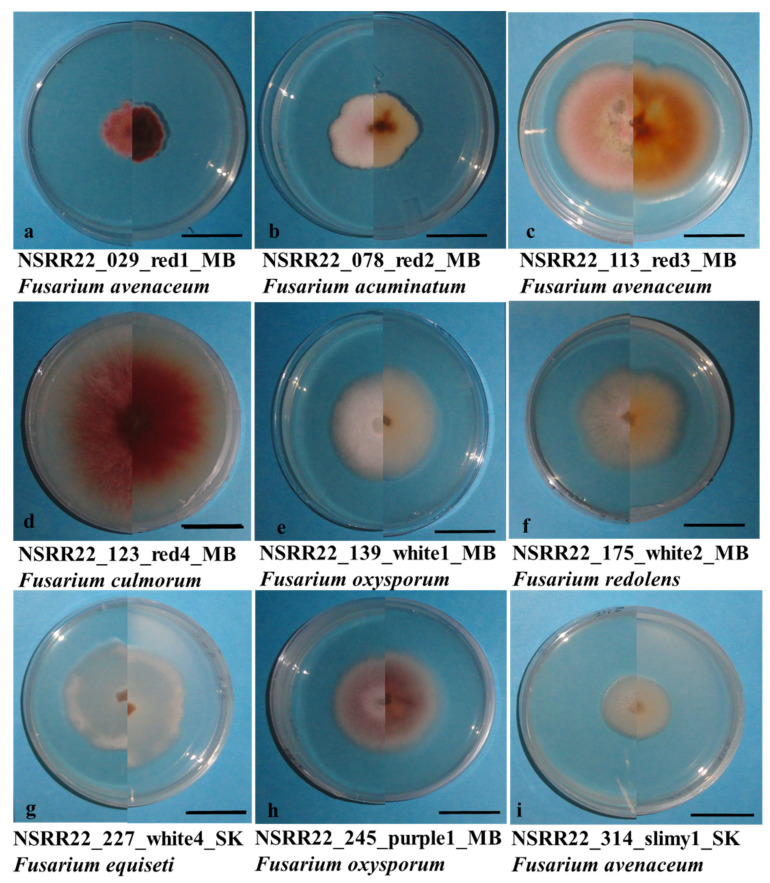
Colony growth on potato dextrose agar of the *Fusarium* isolates (**a**) NSRR22_029, (**b**) NSRR22_078, (**c**) NSRR22_113, (**d**) NSRR22_123, (**e**) NSRR22_139, (**f**) NSRR22_175, (**g**) NSRR22_230, (**h**) NSRR22_245, and (**i**) NSRR22_314, representing the colony morphology subgroups “Red1”, “Red2”, “Red3”, “Red4”, “White1”, “White2”, “White4”, “Purple1”, and “Slimy1”, respectively. The bar in each panel = 1 cm.

**Figure 5 jof-10-00303-f005:**
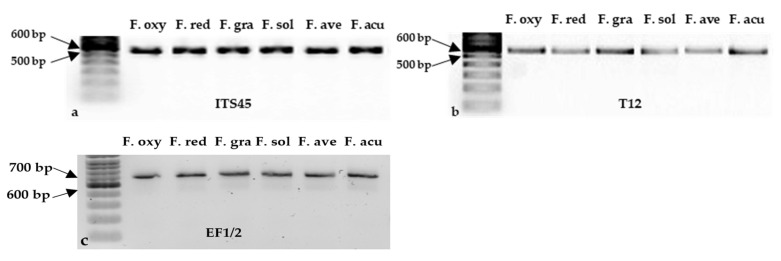
Molecular identification of reference isolates of *Fusarium oxysporum* (F. oxy), *Fusarium redolens* (F. red), *Fusarium graminearum* (F. gra), *Fusarium solani* (F. sol), *Fusarium avenaceum* (F. ave), and *Fusarium acuminatum* (F. acu). Genomic DNA from each isolate was amplified with each of three primer sets: (**a**) ITS4/5 (targeting the ITS region), (**b**) T12 (targeting the beta-tubulin gene), and (**c**) EF1/2 (targeting the *TEF1α* gene), and the products were resolved by electrophoresis on 2% agarose, where a band (500 bp–700 bp) is visible for each isolate. A DNA ladder is included on the left of each panel.

**Figure 6 jof-10-00303-f006:**
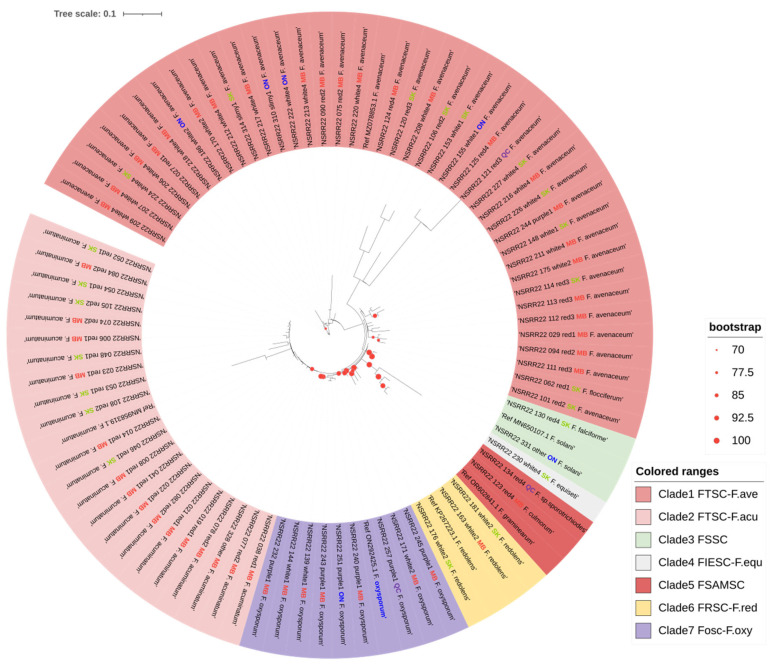
Phylogenetic tree of *Fusarium* isolates identified based on their *TEF1*α sequences. The isolates clustered into seven clades, corresponding to six species complexes: the *Fusarium tricinctum* species complex (FTSC), the *Fusarium solani* species complex (FSSC), the *Fusarium incarnatum-equiseti* species complex (FIESC), the *Fusarium sambucinum* species complex (FSAMSC), the *Fusarium redolens* species complex (FRSC), and the *Fusarium oxysporum* species complex (FOSC). Reference sequences of the six *Fusarium* spp. previously obtained from Manitoba are in bold text. Provinces of origin are color-coded, with Manitoba (MB) in red, Saskatchewan (SK) in green, Ontario (ON) in blue, and Quebec (QC) in purple. Bootstrap values, indicated by circles of varying sizes on the branches, are based on 1000 replicates.

**Table 1 jof-10-00303-t001:** ANOVA table for three germination counts (Count1, Count2, and Count3) recorded at 7, 14, and 21 days after seeding, plant height (Height), root rot severity (RRS), dry shoot weight (Shoot), and dry root weight (Root) of two soybean cultivars, ‘Akras’ and ‘B150Y1′, inoculated under greenhouse conditions with each of 12 fungal isolates representing *Fusarium oxysporum*, *Fusarium redolens*, *Fusarium graminearum*, *Fusarium solani*, *Fusarium avenaceum*, and *Fusarium acuminatum*.

Source of Variance	Df	Mean Square							
		Count1	Count2	Count3	CountAve	Height	RRS	Shoot	Root
F.spp ^1^	12	**10.4 *^,2^**	**4.1 ***	**4.7 ***	**5.5 ***	**53.9 ***	**11 ***	**4.3 ***	**0.2 ***
CV ^1^	1	**128.8 ***	**70.1 ***	**57.3 ***	**82.7 ***	**696.9 ***	**10.7 ***	**175.9 ***	**9.5 ***
Repeat ^1^	1	0.2	1.4	0.3	0.2	1.4	0.0	0.1	0.1
F.spp:CV	12	**3.8 ***	**2.6 ***	**2.0 ***	**2.6 ***	**11.7 ***	**1.0 ***	**0.9 ***	**0.1 ***
F.spp:Repeat	12	1.5	0.8	1.0	0.9	2.8	0.2	0.3	0.0
CV:Repeat	1	1.7	0.9	0.4	0.9	4.5	0.0	0.5	0.1
F.spp:CV:Repeat	12	0.8	0.9	1.0	0.8	3.5	0.2	0.2	0.0
Residuals	208	1.0	0.6	0.6	0.4	4.7	0.1	0.2	0.0

^1^ “F.spp”, “CV”, and “Repeat” refer to the variances from 12 *Fusarium* isolates and non-inoculated controls, two soybean cultivars, and two repeated greenhouse experiments, respectively. ^2^ A bold mean squares denoted with an asterisk (*) indicates that the treatment effect was significant (*p* < 0.001).

**Table 2 jof-10-00303-t002:** ANOVA table for the evaluation of 20 soybean cultivars in response to *Fusarium oxysporum*, *Fusarium redolens*, *Fusarium graminearum*, *Fusarium solani*, *Fusarium avenaceum*, and *Fusarium acuminatum* in greenhouse studies. The estimated traits include three germination counts (Count1, Count2, and Count3) taken at 7, 14, and 21 days after seeding, plant height (Height), root rot severity (RRS), dry shoot weight (Shoot), and dry root weight (Root).

Source of Variance	Df	Mean Square						
		Count1	Count2	Count3	Height	RRS	Shoot	Root
F.spp ^1^	6	**643.2 *^,2^**	**657.1 ***	**709 ***	**2946.8 ***	**227.1 ***	**28.9 ***	**2.12 ***
CV ^1^	19	**28.0 ***	**26.8 ***	**25.8 ***	**56.8 ***	**2.0 ***	**2.2 ***	**0.18 ***
Repeat ^1^	1	0.1	0.1	0.0	0.6	0.3	0.0	0.00
CV:F.spp	114	**3.1 ***	**2.8 ***	**3.1 ***	**15.8 ***	**0.5 ***	**0.2 ***	**0.02 ***
CV:Repeat	19	0.6	0.6	0.6	1.3	0.2	0.0	0.00
F.spp:Repeat	6	1.0	1.7	1.3	0.6	0.5	0.0	0.00
CV:F.spp:Repeat	114	0.8	0.7	0.7	0.9	0.2	0.0	0.00
Residuals	839	1.6	1.6	1.5	2.4	0.3	0.1	0.01

^1^ “F.spp”, “CV”, and “Repeat” refer to the variances from six *Fusarium* species and non-inoculated controls, 20 soybean cultivars, and two repeated greenhouse experiments, respectively. ^2^ Bold mean squares denoted with an asterisk (*) indicate that the treatment effect was significant (*p* < 0.001).

## Data Availability

The analysis of variance, least significant difference analysis of estimated mean in the pathogenicity test, cultivar evaluation in greenhouse trials, and sequence identification in genetic diversity evaluation are available in the main manuscript or as Appendix A.

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
