# Peer review of "Pathogenicity, Host Resistance, and Genetic Diversity of Fusarium Species under Controlled Conditions from Soybean in Canada"

_jof, 2024, doi:10.3390/jof10050303_

Round 1

Reviewer 1 Report

The authors of the manuscript isolated near 1000 strains from soybean rot samples and further identified some strains morphologically similar to Fusarium based on partial sequence of ITS, tubulin and tef1 genes. They tested the pathogenicity of some Fusarium species reported as the major pathogen of soybean rot and evaluated the resistance of 20 soybean cultivars to 6 Fusarium species. These research results provided the basis for further screening of resistant cultivars, but the structure of the manuscript and presentation of the results needs to be adjusted. It would be better to present the results in the order of title of the manuscript: Genetic Diversity, Pathogenicity and Host Resistance. The authors put much to describe the morphology of isolates of Fusarium species. I would suggest authors to refine the relevant result descriptions, because mycelial morphology is only used for preliminary screening of possible Fusarium. Another concern, in pathogenicity test, the authors seems only analyzed the variance of 12 isolates (Tables). What about different Fusarium spp? The figures was not easy to understand.

Page 3, Line 138: ‘one week’, ‘5–day-old’ in Line 141.

Page 4, Line 156: more virulent.  

Page 17, Line 481: ‘suggesting an increase in the aggressiveness of Fusarium spp. in recent decades’, there is no direct evidence to prove that!

Author Response

Reviewer1

Major comments

The authors of the manuscript isolated near 1000 strains from soybean rot samples and further identified some strains morphologically similar to Fusarium based on partial sequence of ITS, tubulin and tef1 genes. They tested the pathogenicity of some Fusarium species reported as the major pathogen of soybean rot and evaluated the resistance of 20 soybean cultivars to 6 Fusarium species. These research results provided the basis for further screening of resistant cultivars, but the structure of the manuscript and presentation of the results needs to be adjusted.

  1. It would be better to present the results in the order of title of the manuscript: Genetic Diversity, Pathogenicity and Host Resistance.

Answer: We appreciate the Reviewer’s suggestions regarding the article structure. However, this study first tested isolate pathogenicity and then evaluated the cultivar reactions; this information was then used to select the more aggressive isolates for evaluation of genetic diversity.  As such, the order presented in the manuscript is the most logical.  However, we agree with the recommendation for consistency with the title. As such, we have modified the title accordingly to match with how the results are presented in the manuscript.

  1. The authors put much to describe the morphology of isolates of Fusarium species. I would suggest authors to refine the relevant result descriptions, because mycelial morphology is only used for preliminary screening of possible Fusarium.

Answer: We appreciate the comment; the morphology description in the results section was reduced accordingly.

  1. Another concern, in pathogenicity test, the authors seems only analyzed the variance of 12 isolates (Tables). What about different Fusarium spp? The figures wasnot easy to understand.

Answer: We thank Reviewer 1 for this question. The least significant difference analysis among the 12 isolates and CK in Fgure 2 indicated that the variance between and among the six Fusarium species is significant. To be consistent with Figure 2 and keep the data in a balanced design, Table 1 accounts for the isolate effect rather than Fusarium species effect. We have added an explanation to the legend in Figure 2 to clarify that the species effect is significant.

Detail comments

Page 3, Line 138: ‘one week’, ‘5–day-old’ in Line 141.

Answer:The growth time for the colonies has been consistently corrected to 7-day-old.

Page 4, Line 156: more virulent.  

Answer: the sentence has been modified as per your suggestion.

Page 17, Line 481: ‘suggesting an increase in the aggressiveness of Fusarium spp. in recent decades’, there is no direct evidence to prove that!

Answer: We appreciate this comment. Reviewer 1 is right. The sentence has been corrected.

Reviewer 2 Report

The authors studied genetic diversity, pathogenicity and host resistance of Fusarium species from soybean in Canada. These findings may help to guide the development of effective measures to mitigate the risk of Fusarium root rot of soybean.

Line 138, Check "Approximately 10 L of wheat grain was soaked in distilled water overnight" , 10 L is right?

Author Response

Reviewer2

Major comments

The authors studied genetic diversity, pathogenicity and host resistance of Fusarium species from soybean in Canada. These findings may help to guide the development of effective measures to mitigate the risk of Fusarium root rot of soybean.

Answer: We thank Reviewer 2 for their positive assessment of this study.

Detail comments

Line 138, Check "Approximately 10 L of wheat grain was soaked in distilled water overnight" , 10 L is right?

Answer: We apologize for the misunderstanding. This sentence described the preparation process for the grain inoculum (i.e., 10 L wheat soaked in autoclavable tub at one time). We have modified the sentence for improved clarity with additional details.

Reviewer 3 Report

This research will contribute immensely to soybean production

The research was thoroughly planned and executed. The results throughout correlated. The only question I had was in the material and method section under 2.1, line 141 where I asked what the size of the mycelial plugs. In section 2.2 line 150, there is an extra space between 25 and the symbol.

Author Response

Reviewer3

Major comments

This research will contribute immensely to soybean production

Answer: We thank Reviewer 3 for their positive assessment of this study.

Detail comments

The research was thoroughly planned and executed. The results throughout correlated. The only question I had was in the material and method section under 2.1, line 141 where I asked what the size of the mycelial plugs. In section 2.2 line 150, there is an extra space between 25 and the symbol.

Answer: We thank the reviewer for this comment. The size of the mycelial plugs is now indicated, and the extra space is also deleted in section 2.2.

Reviewer 4 Report

The data of the paper are suitable to develop its evaluation to multiple resistance and breeding aspects, this is a significant value of the paper, but not discussed or a minor side mention is is in the paper. Therefore the paper should be rethought first, then rewritten. 

This is an interesting paper, the tests seem to be well planned, but there are several problems I cannot suggest the paper to be accepted. I feel a theoretical weakness in the paper, the conclusion and concept is not clearly demonstrated.

Title: I suggest a modification: Genetic Diversity, Pathogenicity and Host Resistance of 2Fusarium Species in Greenhouse Tests from Soybean in Canada

Line 66. The authors listed a high amount of data for multiple Fusarium infection of soybean. The own data prove the same. However, the question never raised, whether a complex resistance to the different Fusarium spp. would be the case. As the authors also made resistance tests on 20 different soybean varieties, the analysis of the data would be highly important in this respect. So, literature should be presented, and when no data exist, the novelty of this research could be expressed better. Do not forget, this is a stone hard breeding problem at the same time.

Line 102. the presence of races of the Fusarium pathogens should also be discussed. F. oxysporum has races, but the population can be mixed with different races or non-specialized pathogen lines, I think, this is an open question. In F. solani there are also races, in this paper no word about them, but they may play a role of infection when race specific resistance genes are present in different varieties.

Line 135. I checked the Chang literature; it seems me that the isolation method resulted pure isolates. A statement will be necessary that these isolates correspond to the monosporic isolate of the classic isolation. Before grain insert wheat grain and give the amount of the grain.

Line 149. Did you use sterilized soil to avoid interactions between your inoculum and the soil microbiome. As this can influence results, there is no word about the evaluation of the pathogenicity tests. There are methods where  other materials sand or perlite are used, and in this case, you can work under sterile conditions with a higher precision. I suppose that the evaluation of the resistance considered the same traits, but it is not sure. Please indicate here the way of the traits, their evaluation. When they were the same, it is enough to describe here and in the resistance part it is enough to mention, that evaluations corresponded (or differed) from the procedure used in the pathogenicity test. I would incorporate the subchapter od data collection. I would omit the greenhouse, as only greenhouse tests were made, it was added to the title, so it is not necessary to mention it again.  Please give an experimental design, replicates, biological replicate and controls. in 2.3. it is enough to mention that the methods and traits were the same ad described at the pathogenicity tests.

Line 155. Experimental design is OK.

Line 171. You evaluated the taproot infection severity. The taproot is only a part of the root system. Why not the total infection severity of the roots was not evaluated. By a q8antitative PCR also the amount of fungal mass could be evaluated.  This is also a possibility to evaluate the disease response. The other thing is that the non-germinated grains (possible plants were not considered (see line 497). This must be done, and the authors remark the this basically influence aggressiveness and resistance ranking. All data should be recalculated by this. This means, for example, when you have five plants and their means, and the control has 10, the missing five should be counted by zero. and not the means for the remaining five plants should be considered. The truth will be shown by the shoot and root mass, because here the rootless plants have no mass, and this should be correlated by the data. of the control. As the author say, this will rewrite the resistance and aggressiveness data (line 498). This should be remarked also in plant height, the only trait that can be left as it is the shoot and root mass. But also, here not the actual numbers should be presented, but the percentages to the control. Of course, the control values should be given in the subheading or by some other way.

3.1. The Figure 1 is too dark, the more infected parts are also dark, so, a differentiation is problematic. I have a problem with germination account. This line composition suggests some development of the curve, even they are independent variables. It would be better to rank the fungal isolates according to their mean aggressiveness. It would be better to use bars, this secures that the aggressiveness gives a ranking. The virulence is normally restricted to the physiological races. In F. oxysporum  and F. solani we have specialized races; this could be a point in the discussion. For the non specialized pathogens like F. graminearum, the disease causing capacity is called aggressiveness. For wheat we know this, but for soybean this is a question, whether a specialization exists or not. This problem is not discussed. It might be that the two isolates from these fungi belong to different races, or differ, therefore, the problem of the possible specialization remains open.  In the plant height data,  the fonts are too small, the designation is hardly readable. The dry matter content would be better in a simplified model,  you can have two different charts for the shoot and root weight data. I do not see a clear conclusion from the data.  For example, are two isolates from s species to characterize the pathogenicity of the given  species. Some correlations between the different symptoms could advise us about difference in aggressiveness and its structure.  In this respect would be interesting which trait was the best to characterize aggressiveness or virulence.

3.2. There are a number of data. However, the question did not raise, whether the resistance to different Fusarium spp. is connected or not. I did not see such way of thinking. The question is, when you select to F. graminearum, will you have automatically higher resistance to the other Fusarium spp, or not.  When yes, one, the most pathogenic species is enough for the breeding work, if not, you have to have parallel screening also for the other Fusarium species. It is also possible, the resistance is common for 2-3 Fusarium spp, and not for the rest. What would like to suggest breeders to do. This is a problem, because in wheat it seems that a common resistance exists, but this is not true for maize for the same fungi. Can you identify common resistance based on the principal component analyses.

4. In line 496 a methodical problem is mentioned, I commented in the methodical part.

Based on this the data should be recalculated. The other point is that I see a correct description of the result, but the evaluation of the data compared to literature in mainly missing.

The experimental part seems to be good, but methodical problems occur that can be improved recalculating the data. We would need an explanation of the results and their possible uses in plant breeding, genetic research etc., as indicated above.  For this reason, the Results and Discussions need to be rewritten.

For this reason, I am not in the position to decide about acceptance. After a major revision this can be the case, but in present form I have only one possibility, to reject the paper.  However, I would reconsider my position when the concept will be clear and consequently treated through the paper.

Author Response

Reviewer4

Major comments

The data of the paper are suitable to develop its evaluation to multiple resistance and breeding aspects, this is a significant value of the paper, but not discussed or a minor side mention is is in the paper. Therefore the paper should be rethought first, then rewritten. 

Answer: We appreciate this comment, which is now reflected in many of the revisions to the manuscript (as indicated in more detail below and on the manuscript itself).

Detail comments

This is an interesting paper, the tests seem to be well planned, but there are several problems I cannot suggest the paper to be accepted. I feel a theoretical weakness in the paper, the conclusion and concept is not clearly demonstrated.

Title: I suggest a modification: Genetic Diversity, Pathogenicity and Host Resistance of 2Fusarium Species in Greenhouse Tests from Soybean in Canada

Answer: The title has been corrected as per our Response to Reviewer 1 and also considering Reviewer 4’s suggestions. Howee

Line 66. The authors listed a high amount of data for multiple Fusarium infection of soybean. The own data prove the same. However, the question never raised, whether a complex resistance to the different Fusarium spp. would be the case. As the authors also made resistance tests on 20 different soybean varieties, the analysis of the data would be highly important in this respect. So, literature should be presented, and when no data exist, the novelty of this research could be expressed better. Do not forget, this is a stone hard breeding problem at the same time.

Answer: We thank Reviewer 4 for their questions regarding broad-spectrum resistance to Fusarium.  This recommendation has been incorporated in the revised manuscript through the revision and expansion of the introduction and discussion section.  The research significance of the two soybean cultivars with tolerance/resistance to different Fusarium spp. is highlighted and placed within the broader context.

Line 102. the presence of races of the Fusarium pathogens should also be discussed. F. oxysporum has races, but the population can be mixed with different races or non-specialized pathogen lines, I think, this is an open question. In F. solani there are also races, in this paper no word about them, but they may play a role of infection when race specific resistance genes are present in different varieties.

Answer: We appreciate this point regarding the further subgrouping of Fusarium spp. We agree with the suggestion and have noted the occurrence of formae speciales and races both in the revised introduction and discussion sections of the manuscript. 

Line 135. I checked the Chang literature; it seems me that the isolation method resulted pure isolates. A statement will be necessary that these isolates correspond to the monosporic isolate of the classic isolation. Before grain insert wheat grain and give the amount of the grain.

Answer: We thank Reviewer 4 for this useful suggestion regarding the methods section.  The single-spore isolation method applied for the Fusarium spp. is now mentioned, and the initial wheat grain volume and final volume of the inoculum powder are described.

Line 149. Did you use sterilized soil to avoid interactions between your inoculum and the soil microbiome. As this can influence results, there is no word about the evaluation of the pathogenicity tests. There are methods where  other materials sand or perlite are used, and in this case, you can work under sterile conditions with a higher precision. I suppose that the evaluation of the resistance considered the same traits, but it is not sure. Please indicate here the way of the traits, their evaluation. When they were the same, it is enough to describe here and in the resistance part it is enough to mention, that evaluations corresponded (or differed) from the procedure used in the pathogenicity test. I would incorporate the subchapter od data collection. I would omit the greenhouse, as only greenhouse tests were made, it was added to the title, so it is not necessary to mention it again.  Please give an experimental design, replicates, biological replicate and controls. in 2.3. it is enough to mention that the methods and traits were the same ad described at the pathogenicity tests.

Answer. We appreciate this point. The sterilization method for the potting medium is now noted.  On the other hand, the data collection for the two greenhouse experiment was exactly the same, and this is also now noted in the methods.

Line 155. Experimental design is OK.

Answer: We thank Reviewer 4 for this confirmation.

Line 171. You evaluated the taproot infection severity. The taproot is only a part of the root system. Why not the total infection severity of the roots was not evaluated. By a q8antitative PCR also the amount of fungal mass could be evaluated.  This is also a possibility to evaluate the disease response. The other thing is that the non-germinated grains (possible plants were not considered (see line 497). This must be done, and the authors remark the this basically influence aggressiveness and resistance ranking. All data should be recalculated by this. This means, for example, when you have five plants and their means, and the control has 10, the missing five should be counted by zero. and not the means for the remaining five plants should be considered. The truth will be shown by the shoot and root mass, because here the rootless plants have no mass, and this should be correlated by the data. of the control. As the author say, this will rewrite the resistance and aggressiveness data (line 498). This should be remarked also in plant height, the only trait that can be left as it is the shoot and root mass. But also, here not the actual numbers should be presented, but the percentages to the control. Of course, the control values should be given in the subheading or by some other way.

Answer: We thank the reviewer for this question. The evaluation of severity considers disease development on the entire root system. Our disease severity scale is not only based on symptoms on the taproot, but also assesses the fibrous roots, stem and above ground tissues. We have provided more details on the disease severity scale in the materials and methods to avoid potential misunderstandings.

Our study originally calculated the estimated means for plant height and disease severity based on two methods: (1) average of existing plants (Height1 and DS1); (2) sum of all pseudoreplicates within one cup (treating non-germinated seeds as dead plants, thus height is 0 cm and DS is 4) divided by the total seed number of a cup (Height2 and DS2). Based on your suggestion, the first calculation method has been removed and only the second method is used to interpret the results. All the data have been re-calculated and all of the graphs have been re-made.

3.1. The Figure 1 is too dark, the more infected parts are also dark, so, a differentiation is problematic. I have a problem with germination account. This line composition suggests some development of the curve, even they are independent variables. It would be better to rank the fungal isolates according to their mean aggressiveness. It would be better to use bars, this secures that the aggressiveness gives a ranking. The virulence is normally restricted to the physiological races. In F. oxysporum  and F. solani we have specialized races; this could be a point in the discussion. For the non specialized pathogens like F. graminearum, the disease causing capacity is called aggressiveness. For wheat we know this, but for soybean this is a question, whether a specialization exists or not. This problem is not discussed. It might be that the two isolates from these fungi belong to different races, or differ, therefore, the problem of the possible specialization remains open.  In the plant height data,  the fonts are too small, the designation is hardly readable. The dry matter content would be better in a simplified model,  you can have two different charts for the shoot and root weight data. I do not see a clear conclusion from the data.  For example, are two isolates from s species to characterize the pathogenicity of the given  species. Some correlations between the different symptoms could advise us about difference in aggressiveness and its structure.  In this respect would be interesting which trait was the best to characterize aggressiveness or virulence.

Answer: We thank this reviewer for their insights into Fusarium spp. More detailed descriptions of the root symptoms were added to the legend of Figure 1. The graphs in Figure 2 were remade following the reviewer’s suggestion. The questions of aggressiveness and host specialization are now addressed in the revised discussion.s

3.2. There are a number of data. However, the question did not raise, whether the resistance to different Fusarium spp. is connected or not. I did not see such way of thinking. The question is, when you select to F. graminearum, will you have automatically higher resistance to the other Fusarium spp, or not.  When yes, one, the most pathogenic species is enough for the breeding work, if not, you have to have parallel screening also for the other Fusarium species. It is also possible, the resistance is common for 2-3 Fusarium spp, and not for the rest. What would like to suggest breeders to do. This is a problem, because in wheat it seems that a common resistance exists, but this is not true for maize for the same fungi. Can you identify common resistance based on the principal component analyses.

Answer. We appreciate this suggestion regarding common resistance in soybean to multiple Fusarium spp. The broad-spectrum resistance in the soybean cultivars observed in this study is now addressed in more detail in the discussion section.

  1. In line 496 a methodical problem is mentioned, I commented in the methodical part.

Based on this the data should be recalculated. The other point is that I see a correct description of the result, but the evaluation of the data compared to literature in mainly missing.

Answer: The data were re-calculated following Reviewer 4’s suggestion.

The experimental part seems to be good, but methodical problems occur that can be improved recalculating the data. We would need an explanation of the results and their possible uses in plant breeding, genetic research etc., as indicated above.  For this reason, the Results and Discussions need to be rewritten.

For this reason, I am not in the position to decide about acceptance. After a major revision this can be the case, but in present form I have only one possibility, to reject the paper.  However, I would reconsider my position when the concept will be clear and consequently treated through the paper.

Answer: We sincerely thank Reviewer 4 for all of their constructive criticism, which has helped to improve this manuscript.  Their impressive knowledge and suggestions have been most helpful.

Round 2

Reviewer 1 Report

My comments have been fully addressed in the revised manuscript.

My comments have been fully addressed in the revised manuscript.

Reviewer 4 Report

Line 101. I should mention mycelium also as infecting agent as Takegami and Sasai found it in Japan in 1970.

Figure 1. is now good, you can see the differences.

Line 210. As I have looked the dictionary, the taproot meant the main root.  In several plants you have a rich root system. The question is whether your evaluation relates to the main root or to all roots you have. Maybe my dictionary is bad. Anyway, a clarification is needed. It would be better to say (when this is the truth) that the whole root system was considered. This would be clear. The other thing is that F. oxysporum may also cause a general trachea mycosis that stops the water and nutrient transport, and the plants die in a very short time. Did you check this symptom?

Line 305 in the new version. I agree, the non-germinated grains received o for plant height or root length. But it is not clear, how root rot was evaluated. I would say, they should receive the maximum value. Otherwise, when you have 5 plants from the 10 grains, and they are healthy, all will receive zero, but shoot mass or root mass will be half of the control at the same mean root rot. This should be clarified. When I do not know wheat the data mean, it is not easy to find out whet is the situation. When disease severity means root rot severity, I would use this expression as RRS as abbreviation for example.

Line 332. Does refer disease severity root rot. This should be clarified. When this is true, tits evaluation as I indicated above is a highly important point to explain.

So, I should say that the paper become much better.  However, I am not in the position to evaluate the results until I do not know, how these numbers were counted and what they exactly mean.

By this way, I stop the evaluation and wait for the new version.

For this reason, I cannot decide about accept or reject the paper, both possibilities are open, in this version closed to acceptation.

Line 101. I should mention mycelium also as infecting agent as Takegami and Sasai found it in Japan in 1970.

Figure 1. is now good, you can see the differences.

Line 210. As I have looked the dictionary, the taproot meant the main root.  In several plants you have a rich root system. The question is whether your evaluation relates to the main root or to all roots you have. Maybe my dictionary is bad. Anyway, a clarification is needed. It would be better to say (when this is the truth) that the whole root system was considered. This would be clear. The other thing is that F. oxysporum may also cause a general trachea mycosis that stops the water and nutrient transport, and the plants die in a very short time. Did you check this symptom?

Line 305 in the new version. I agree, the non-germinated grains received o for plant height or root length. But it is not clear, how root rot was evaluated. I would say, they should receive the maximum value. Otherwise, when you have 5 plants from the 10 grains, and they are healthy, all will receive zero, but shoot mass or root mass will be half of the control at the same mean root rot. This should be clarified. When I do not know wheat the data mean, it is not easy to find out whet is the situation. When disease severity means root rot severity, I would use this expression as RRS as abbreviation for example.

Line 332. Does refer disease severity root rot. This should be clarified. When this is true, tits evaluation as I indicated above is a highly important point to explain.

So, I should say that the paper become much better.  However, I am not in the position to evaluate the results until I do not know, how these numbers were counted and what they exactly mean.

By this way, I stop the evaluation and wait for the new version.

For this reason, I cannot decide about accept or reject the paper, both possibilities are open, in this version closed to acceptation.

Author Response

Line 101. I should mention mycelium also as infecting agent as Takegami and Sasai found it in Japan in 1970.

Answer: Thank you for your suggestion. We have added a statement indicating that mycelium may also cause infection.  

Figure 1. is now good, you can see the differences.

Answer: We appreciate your comment. Thank you.

Line 210. As I have looked the dictionary, the taproot meant the main root.  In several plants you have a rich root system. The question is whether your evaluation relates to the main root or to all roots you have. Maybe my dictionary is bad. Anyway, a clarification is needed. It would be better to say (when this is the truth) that the whole root system was considered. This would be clear. The other thing is that F. oxysporum may also cause a general trachea mycosis that stops the water and nutrient transport, and the plants die in a very short time. Did you check this symptom?

Answer: Thank you for your suggestions. The description of the root rot severity scale and related information has been modified for clarity as per your feedback.

Line 305 in the new version. I agree, the non-germinated grains received o for plant height or root length. But it is not clear, how root rot was evaluated. I would say, they should receive the maximum value. Otherwise, when you have 5 plants from the 10 grains, and they are healthy, all will receive zero, but shoot mass or root mass will be half of the control at the same mean root rot. This should be clarified. When I do not know wheat the data mean, it is not easy to find out whet is the situation. When disease severity means root rot severity, I would use this expression as RRS as abbreviation for example.

Answer: Thank you for your suggestion. The details regarding the calculation of all traits in the current paper have been re-written in the materials and methods section for improved clarity. The non-germinating plants were recorded as having a height = 0 and a root rot severity = 4. Both disease severity and root rot severity are expressed as “RRS”.

Line 332. Does refer disease severity root rot. This should be clarified. When this is true, tits evaluation as I indicated above is a highly important point to explain.

Answer: Thank you for your suggestion. The disease severity refers to the root rot severity in the current study. More explanation has been added to the materials and methods section.

So, I should say that the paper become much better.  However, I am not in the position to evaluate the results until I do not know, how these numbers were counted and what they exactly mean.

By this way, I stop the evaluation and wait for the new version.

For this reason, I cannot decide about accept or reject the paper, both possibilities are open, in this version closed to acceptation.

Answer: Thank you for your comments.